# Biological Activities Underlying the Cardiovascular Benefits of Olive Oil Polyphenols: Focus on Antioxidant, Anti-Inflammatory, and Anti-Atherogenic Effects

**DOI:** 10.3390/ijms262211165

**Published:** 2025-11-19

**Authors:** Kaoutar Boumezough, Mehdi Alami, Tamas Fulop, Nada Zoubdane, Ikram Salih, Mhamed Ramchoun, Abdelouahed Khalil, Hicham Berrougui

**Affiliations:** 1Department of Biology, Polydisciplinary Faculty, Sultan Moulay Sliman University, Beni Mellal 23020, Morocco; kaoutarrosa1@gmail.com (K.B.); mhamed.ramchoun@gmail.com (M.R.); 2Department of Medicine, Geriatrics Service, Faculty of Medicine and Health Sciences, University of Sherbrooke, Sherbrooke, QC J1H 4N4, Canada; mehdi.alami@usherbrooke.ca (M.A.); tamas.fulop@usherbrooke.ca (T.F.); zoubdane.nada@usherbrooke.ca (N.Z.); Ikram.Salih@usherbrooke.ca (I.S.);; 3Research Laboratory in Oral Biology and Biotechnology, Faculty of Dental Medicine, Mohammed V University in Rabat, BP 6212, Rabat 10000, Morocco

**Keywords:** polyphenols, olive oil, antioxidant, anti-inflammatory, anti-atherogenic

## Abstract

Extra virgin olive oil (EVOO) polyphenols are recognized for their beneficial effects on human health, yet how their concentration shapes biological outcomes remains insufficiently explored. While a daily intake of 25 mL EVOO is generally regarded as beneficial for cardiovascular protection, the high-phenolic EVOO examined in this study contains markedly higher levels of polyphenols than most EVOOs reported previously. This suggests that oils richer in polyphenols may exert distinct biological effects. To investigate this, we compared extracts from a standard EVOO and a naturally high-phenolic EVOO, along with their key phenolic compounds, hydroxytyrosol (HT) and tyrosol (Tyr). Antioxidant effects were assessed by quantifying intracellular reactive oxygen species (ROS) and lipid peroxidation. Anti-inflammatory activity was evaluated in THP-1-derived macrophages stimulated with LPS by analyzing inflammatory surface markers’ expression, cytokines’ production, and the NLRP3-inflammasome pathway. Atheroprotective potential was investigated by measuring cholesterol efflux in J774 macrophages. Both EVOO polyphenols extracts and (HT and Tyr) significantly reduced ROS and lipid peroxidation. High phenolic EVOO extract (EVOOPE+) displayed superior antioxidant activity at lower concentrations, while standard EVOO phenolic extract (EVOOPE) showed more consistent effects across doses. Both extracts favored an anti-inflammatory macrophage phenotype, as indicated by increased CD163 and IL-10 expression and reduced CD86, IFN-α, and NLRP3. Moreover, all treatments enhanced cholesterol efflux in a dose-dependent manner, with EVOOPE+ and HT producing the strongest effects. Collectively, these results highlight the capacity of EVOO polyphenols to modulate, through key bioactivity mechanisms, cardioprotective effects and emphasize the importance of polyphenols concentration in their biological efficacy.

## 1. Introduction

As cardiovascular diseases (CVDs) have emerged as the leading global health burden, there is growing interest in lifestyle strategies that foster long-term health. Diet, once viewed primarily as a means of sustenance, is now recognized as a crucial factor in disease prevention and physiological resilience. Among the various dietary models explored for their protective effects, the Mediterranean diet has consistently been associated with enhanced longevity, improved metabolic regulation, and a lower risk of chronic illnesses [1].

Extra virgin olive oil (EVOO) is a fundamental element of the Mediterranean diet, highly regarded not only for its distinctive flavor and culinary adaptability but also for its unique composition.

Olive oils are classified according to their extraction method, acidity, and sensory characteristics into several categories: EVOO, virgin olive oil (VOO), and refined olive oil. EVOO stands as the pinnacle of quality, mandated to be obtained solely through mechanical pressing or centrifugation without the use of heat or chemical solvents. This careful processing preserves its chemical integrity, resulting in a free acidity of less than 0.8% and the absence of any sensory defects, yielding an oil with superior, complex flavor and aroma. VOO, while also unrefined, permits a slightly higher free acidity level of up to 2.0% and may exhibit minor sensory imperfections. In stark contrast, refined olive oil is produced through chemical and thermal treatments applied to virgin oils of poor quality, processes which strip away many of the oil’s natural compounds, leading to a product with significantly reduced nutritional and organoleptic value [2,3,4].

From a chemical perspective, EVOO’s reputation is substantiated by its complex composition. It is fundamentally a mixture of a saponifiable fraction, which constitutes 98–99% of its weight and is predominantly composed of triglycerides rich in oleic acid, and a minor unsaponifiable fraction (0.5–1.5%). Although small in quantity, this unsaponifiable fraction is disproportionately responsible for the oil’s bioactivity and identity. It is a repository of diverse micronutrients, including tocopherols (notably vitamin E), phytosterols, pigments like chlorophylls and carotenoids, aliphatic alcohols, and a vast array of volatile organic compounds that define its aroma. Crucially, this fraction is also rich in phenolic compounds (PCs), a class of powerful antioxidant and anti-inflammatory molecules now widely regarded as the principal bioactive constituents responsible for most of EVOO’s documented health-promoting effects, including cardioprotective and neuroprotective activities. Thus, the superior status of EVOO is not merely sensory but is fundamentally rooted in its unique and preserved phytochemical matrix [2,3,4].

Polyphenols constitute a vast and structurally diverse family of phytochemicals, universally characterized by the presence of multiple phenolic units, aromatic rings bearing one or more hydroxyl functional groups. As ubiquitous secondary metabolites in the plant kingdom, they represent one of the most extensive groups of known bioactive compounds, with over 8000 unique structures identified to date, several hundred of which are present in the human diet. Principal dietary sources encompass a wide range of plant-based foods, including fruits, vegetables, cereals, nuts, seeds, tea, and coffee [5]. Within the context of olive oil, the phenolic profile is particularly rich, with more than 36 distinct phenolic compounds (PCs) identified, exhibiting a broad concentration range of 50 to 1000 mg/kg. Among these, the phenolic alcohols tyrosol (Tyr) and hydroxytyrosol (HT) are quantitatively predominant, collectively constituting a substantial portion of the total phenolic content, with average relative abundances of approximately 46% and 36%, respectively.

The concentration and composition of these compounds are not static but are influenced by a complex interplay of agronomic, technological, and storage factors, including the olive cultivar, growing conditions, fruit maturity at harvest, extraction techniques, and post-production storage. Structurally, olive oil phenolic compounds (OOPCs) can be systematically classified into several major groups: phenolic acids (e.g., p-coumaric, gallic, and caffeic acids), flavonoids (e.g., luteolin and apigenin derivatives), lignans (e.g., pinoresinol and 1-acetoxypinoresinol), secoiridoids (which are particularly characteristic of the *Olea europaea* species and include oleuropein and ligstroside along with their aglycones and derivatives), phenolic alcohols (such as Tyr and HT), and isochromans [3,4].

Growing research supports the role of OOPCs in preserving cardiometabolic health, through their ability to mitigate oxidative damage, regulate inflammatory processes, and enhance metabolic balance. Their influence extends across lipid metabolism, vascular health, and cellular defense mechanisms, collectively shaping an individual’s cardiometabolic risk profile.

Both clinical and experimental evidence demonstrate that habitual intake of polyphenols-rich EVOO improves lipid parameters, boosts antioxidant defenses, and lowers inflammatory markers. This underscores their potential as natural, complementary agents in the prevention and management of chronic metabolic diseases [3,6]. Altogether, these findings emphasize the value of OOPCs as effective contributors to cardiometabolic wellness. Incorporating these compounds into daily nutritional practices not only honors time-tested dietary traditions but also offers a scientifically validated approach to mitigating the growing impact of lifestyle-associated disorders [3]. EVOO polyphenols are recognized for their beneficial effects on human health, yet how their concentration shapes biological outcomes remains insufficiently explored. This work aims to investigate the biological activities underlying the cardiovascular benefits of a high–phenolic–content EVOO compared with a standard EVOO, focusing particularly on its main phenolic compounds, Tyr and HT. Special attention is given to their antioxidant, anti-inflammatory, and anti-atherogenic effects.

## 2. Results

### 2.1. Total Phenolic Content and Fatty Acid Composition

The total phenolic content of the analyzed extracts, quantified using a gallic acid standard curve, is presented in Table 1. EVOOPE+ exhibited a markedly higher total phenolic concentration than EVOOPE. As previously published by our team [7], EVOO+ was particularly enriched in HT compared to Tyr, with concentrations of 233.6 mg/kg and 123.1 mg/kg of oil, respectively. In contrast, regular EVOO contained substantially lower levels of both HT and Tyr, at 7.8 mg/kg and 6.3 mg/kg of oil, respectively, highlighting the enhanced phenolic profile of EVOO+.

The fatty acid (FA) composition of the three analyzed oils is presented in Appendix A. The FA profile reveals that EVOO+ contains a substantially higher proportion of oleic acid—an anti-inflammatory monounsaturated fatty acid—by 18.7% compared with EVOO. Conversely, EVOO+ exhibits markedly lower levels of palmitic acid, a pro-inflammatory saturated fatty acid, by 30.21%, relative to EVOO. Furthermore, the linoleic acid (ω-6) to α-linolenic acid (ω-3) ratio is reduced in EVOO+ (16.67:1) compared with EVOO (25.95:1). This lower ω-6/ω-3 ratio suggests that linoleic acid in EVOO+ exhibits a reduced pro-inflammatory potential.

### 2.2. Cell Viability Measurement

As a first approach, we evaluated the effect of increasing concentrations of polyphenol extracts and their main compounds on the viability of THP-1 cells using the MTT assay. The results showed that none of the treatments significantly affected cell viability. This observation suggests that the doses used are safe for the cells and suitable for subsequent functional assays, such as investigations of antioxidant, anti-inflammatory, or anti-atherogenic effects (Table 2).

### 2.3. Antioxidant Activity

Our findings demonstrate that both extracts, along with their principal phenolic constituents HT and Tyr, exhibit significant antioxidant activity, as evidenced by their ability to lower intracellular ROS levels and inhibit lipid peroxidation. Data from the DCFH-DA assay (Figure 1) revealed that EVOOPE (Figure 1A) and EVOOPE+ (Figure 1B) significantly decreased ROS production in J774 macrophages subjected to oxidative stress. Notably, EVOOPE+ demonstrated greater efficacy at lower concentrations (25 and 50 µg/mL), which may be attributed to its higher total phenolic content. However, its effect diminished at higher concentrations (100 and 150 µg/mL), implying a potential biphasic dose–response or a shift to pro-oxidant activity at elevated doses. A similar pattern was observed for pure Tyr (Figure 1C) and HT (Figure 1D). These molecules significantly reduce ROS formation in a dose-dependent manner, with HT exhibiting a more potent effect. Notably, at the highest concentration tested (150 µg/mL), the cytoprotective effect of HT was reduced, yielding an efficacy comparable to that of Tyr at the same concentration (Appendix A).

As illustrated in Figure 2, polyphenol extracts (Figure 2A,B) as well as Tyr and HT (Figure 2C,D). Significantly reduced lipid peroxidation. Notably, at lower concentrations, EVOOPE+ and its most potent component, HT, demonstrated superior activity compared to EVOOPE and Tyr (Figure 2A). However, mirroring the results of the ROS assay, the protective effect of EVOOPE+ diminished at higher concentrations. This contrasts with EVOOPE, which exhibited a more consistent, dose-dependent response. This observation suggests that despite a lower total phenolic content, EVOOPE may possess a more balanced or stable phytochemical composition that preserves its antioxidant capacity at elevated doses. A parallel trend was observed with the pure compounds; while both HT and Tyr effectively reduced lipid peroxidation, the potency of HT slightly declined at the highest concentration (150 µM), whereas Tyr maintained a stable effect. This further indicates a potential concentration-dependent shift in redox behavior, particularly for HT, which may transition from an antioxidant to a less effective or pro-oxidant state at high doses Appendix A.

### 2.4. Anti-Inflammatory Activity

To delineate the immunomodulatory potential of OOPEs, we investigated the effects of EVOOPE and EVOOPE+ on key inflammatory and macrophage polarization markers in LPS-stimulated THP-1-derived macrophages. We further compared these findings with the effects of the two principal phenolic compounds, HT and Tyr, to identify potential molecular drivers of the observed responses.

CD86, a surface marker associated with macrophage maturation and antigen-presenting activity, was significantly upregulated following LPS stimulation. Treatment with both EVOOPE (Figure 3A) and EVOOPE+ (Figure 3B) led to a marked reduction in CD86 expression, with EVOOPE+ inducing a more pronounced suppression (*p* < 0.001). In contrast, the expression of CD163, a canonical marker of anti-inflammatory M2 macrophages, was significantly upregulated by both extracts (Figure 3C,D). EVOOPE+ consistently induced higher CD163 expression than EVOOPE at 50 and 100 µg/mL (Appendix A), indicating a more potent shift toward an immunoresolving M_2_-like phenotype.

Importantly, HT and Tyr also significantly modulated the expression of these markers across the same range of statistical significance (*p* < 0.05 to *p* < 0.001), by decreasing CD86 expression (Figure 4A,B) and increasing CD163 expression (Figure 4C,D). HT demonstrates a relatively more potent effect (Appendix A), further supporting its role as an active contributor to the immunomodulatory effects of the extracts.

The anti-inflammatory potential of the extracts was further demonstrated by their ability to modulate IL-10 secretion. While LPS challenge induced a marked downregulation of this critical anti-inflammatory cytokine, co-incubation with either EVOOPE or EVOOPE+ effectively counteracted this effect. Quantitative analysis revealed that both extracts significantly and dose-dependently restored IL-10 expression (*p* < 0.05 to *p* < 0.001; Figure 5A,B). Consistent with its overall bioactivity profile, EVOOPE+ promoted a superior restorative effect at 25 µg/mL compared to EVOOPE (Appendix A), underscoring its more potent anti-inflammatory properties.

Consistent with its pro-inflammatory role, IFN-α levels were markedly elevated in LPS-treated cells (*p* < 0.001 vs. control). Treatment with either extract significantly reduced IFN-α production (Figure 6A,B). EVOOPE+ (Figure 6A) demonstrated a more substantial suppressive effect, reaching statistical significance from 25 µg/mL onward (*p* < 0.001), and outperforming EVOOPE (Appendix A). These findings suggest that EVOOPE+ more effectively inhibits interferon-driven inflammatory pathways. Similarly, LPS significantly induced IL-6 expression (*p* < 0.001), and both extracts reduced IL-6 levels across tested concentrations (*p* < 0.05 to *p* < 0.001; Figure 6C,D). However, no significant differences were observed between EVOOPE and EVOOPE+, indicating comparable efficacy in modulating IL-6–mediated inflammation (Appendix A). In contrast, only a modest reduction in IL-1β levels was observed following extract treatment (Figure 6E,F), with a nonsignificant trend suggesting a slightly greater effect of EVOOPE+ (Appendix A).

As illustrated in Figure 7, stimulation with LPS resulted in a marked increase in NLRP3 expression in THP-1-derived macrophages, confirming the effective activation of the inflammasome pathway. Both EVOOPE+ (Figure 7A) and EVOOPE (Figure 7B) significantly reduced NLRP3 levels.

Intriguingly, the upregulation of specific immunomodulatory markers, including CD163 and IL-10, alongside the suppression of the NLRP3 inflammasome, exhibited a biphasic response to EVOOPE+. Maximum effects were achieved at intermediate concentrations, with a notable decline in efficacy at the highest dose (150 µg/mL). This non-linear pattern implies that the anti-inflammatory activity of the extract may be subject to dose-limiting effects, potentially arising from pro-oxidant behavior or the triggering of adaptive cellular regulatory pathways at high concentrations.

To elucidate the bioactive components underlying the observed effects of the extracts, we examined the impact of HT and Tyr on macrophages. Both compounds significantly increased IL-10 expression (Figure 8(A1,B1)). Tyr alone showed a modest, concentration-dependent tendency to reduce IL-1β expression (Figure 8(A2,B2)). Both HT and Tyr strongly suppressed IFN-α production (Figure 8(A3,B3); *p* < 0.001), highlighting their ability to interfere with type I interferon signaling. In contrast, IL-6 expression remained unchanged after treatment with either compound (Figure 8(A4,B4)). Finally, both phenolics reduced NLRP3 levels (Figure 8(A5,B5)), with HT exerting a slightly stronger effect, consistent with its role in inhibiting inflammasome activation (Appendix A).

### 2.5. Cholesterol Efflux Measurement in J774 Macrophages

The results showed that OOPEs (Figure 9A,B), Tyr, and HT (Figure 9C,D) significantly enhanced cholesterol efflux in macrophages in a dose-dependent manner. Among the extracts tested, EVOOPE+ exhibited a greater effect than EVOOPE at all concentrations tested (Appendix A). Furthermore, HT induced a higher level of cholesterol efflux compared to Tyr (Appendix A).

## 3. Discussion

### 3.1. Antioxidant Activity

The mechanisms underlying the action of polyphenols, particularly their antioxidant properties and interaction with endogenous defense systems, have long attracted scientific interest. Although many aspects remain to be fully elucidated, substantial knowledge has already been gained regarding their antioxidant effects and their role in disease prevention [3,8]. Mechanistically, OOPCs are well-known for their antioxidant properties, primarily through radical scavenging and metal chelation [3,9,10]. However, there is a body of scientific literature that supports the idea that polyphenols are generally antioxidants that can act as pro-oxidants under certain conditions [8,10], and highlight their biphasic redox behavior, where under certain conditions, such as high concentrations, elevated pH, and the presence of transition metal ions like iron or copper, these same compounds can paradoxically exhibit pro-oxidant effects [9,11,12,13]. Our results align with this duality, revealing a dose-dependent antioxidant activity of OOPCs at lower concentrations, contrasted by a loss of efficacy or even a potential pro-oxidant trend at higher doses. This phenomenon is not unique to OOPCs; other dietary polyphenols, including flavonoids like quercetin and hydroxycinnamic acids, have shown similar redox cycling behaviors, particularly in the presence of oxygen and redox-active metals, leading to ROS generation and oxidative damage to lipids, proteins, and DNA [9,14]. Notably, this pro-oxidant potential is also supported by in vivo findings. For instance, Kouka et al. (2020) observed that while olive oil enhanced antioxidant defenses in specific tissues (e.g., blood and brain), it simultaneously induced oxidative stress in others, such as the liver and heart [15], suggesting a tissue-specific response [12,15]. Similarly, high doses of HT, a primary phenolic compound in olive oil, have been reported to trigger systemic pro-oxidant effects and glutathione depletion in rats [16]. These findings underscore the importance of dose and context in determining the biological outcome of polyphenol exposure. While polyphenol-rich supplements are often marketed for their health-promoting potential, the evidence points to a double-edged sword effect: low-to-moderate doses may offer protection, but high doses may disrupt redox homeostasis and interfere with endogenous antioxidant defenses [8,11]. Therefore, the interpretation of polyphenol bioactivity, particularly in vivo, must consider this complex redox behavior, which may vary depending on the concentration, chemical structure, tissue distribution, and biological context.

### 3.2. Anti-Inflammatory Activity

This study demonstrates that phenolic extracts from extra virgin olive oil exert pronounced anti-inflammatory effects in LPS-stimulated THP-1-derived macrophages. These effects include a reduction in pro-inflammatory mediators such as IFN-α, CD86, IL-6, and NLRP3, along with an increase in anti-inflammatory and M2-polarization markers like IL-10 and CD163. Given the central role of macrophage plasticity in the initiation and progression of atherosclerosis, these results are particularly relevant for the prevention and management of cardiovascular diseases (CVDs).

The strong upregulation of IL-10, a pivotal immunoregulatory cytokine, by EVOOPE+ supports a shift toward inflammation resolution. IL-10 not only inhibits classical pro-inflammatory signaling cascades such as NF-κb but also modulates antigen-presenting cell function and promotes the differentiation of regulatory macrophages and T cells. Alongside IL-10, the enhanced expression of CD163 further reflects the promotion of an M2-like anti-inflammatory phenotype, known to reduce plaque instability and facilitate tissue repair in atherosclerotic lesions [17,18]. EVOOPE+ also markedly reduced IFN-α expression, a cytokine central to the type I interferon response that sustains chronic inflammation and vascular damage in CVDs. The inhibition of NLRP3 expression underscores the potential of EVOOPE+ to reduce inflammasome priming. Although IL-1β levels showed only minor changes, this may be due to the absence of a second activation signal in the experimental design. Nevertheless, NLRP3 inhibition is significant, as its chronic activation has been directly linked to endothelial dysfunction and plaque rupture [19]. HT showed potent NLRP3 inhibition, in line with its known capacity to interfere with NF-κB and oxidative stress-dependent priming of inflammasomes. Interestingly, while Tyr and HT alone reproduced several extract effects (e.g., IL-10 upregulation, IFN-α suppression), they failed to fully match the potency or breadth of the complete EVOOPE+ extract. This observation supports the contribution of other phenolic compounds such as oleuropein, ligstroside derivatives, flavonoids (e.g., luteolin, apigenin), and lignans (e.g., pinoresinol), to the observed bioactivity. These compounds have been shown to exert additive or synergistic effects on macrophage behavior, cytokine release, and oxidative stress through interactions with the MAPK, JAK/STAT, and PI3K/Akt pathways [20]. For example, secoiridoids like oleocanthal and oleacein are reported to inhibit iNOS and COX-2, while flavonoids modulate the phosphorylation of key inflammatory transcription factors [21,22]. The presence of such compounds in EVOOPE+, at higher concentrations than in EVOOPE, may explain the broader and more consistent anti-inflammatory responses observed. Moreover, the suppression of CD86, an antigen presentation marker linked to T-cell priming and chronic inflammation, suggests that EVOOPE+ may dampen the interface between innate and adaptive immunity, a relevant target in atherosclerosis and metabolic syndrome [23]. As inflammation resolution relies not only on cytokine rebalancing but also on macrophage reprogramming and antigen tolerance, the capacity of EVOO phenolics to modulate surface markers and signalling interfaces is particularly promising.

The non-linear dose–response observed, where high concentrations (150 µg/mL) occasionally showed diminished effects, could reflect saturation or activation of compensatory pathways. Such hormetic behaviour is well-recognised for polyphenols and is attributed to their biphasic regulation of redox-sensitive signalling, mitochondrial dynamics, and ER stress [14,24]. Importantly, this emphasises the need for precise dose optimization when formulating phenolic-rich interventions for immune and vascular health.

Taken together, our findings demonstrate that EVOOPE+ exhibits multifaceted immunomodulatory activity through the selective regulation of key inflammatory markers, suppression of the type I interferon and NLRP3 pathways, and promotion of anti-inflammatory reprogramming in macrophages. These properties make it a strong candidate for dietary or adjunct therapeutic strategies targeting chronic low-grade inflammation in cardiovascular diseases. The contribution of individual compounds, such as HT and Tyr, is significant; however, the data also underscore the importance of the complete phenolic profile, including minor yet bioactive compounds that act synergistically to enhance efficacy and specificity.

### 3.3. Anti-Atherogenic Activity

The present findings provide compelling evidence that OOPCs play a significant role in promoting cholesterol efflux from macrophages, a key component of reverse cholesterol transport (RCT). This mechanism is central to the early prevention of atherogenesis, as the accumulation of cholesterol-loaded macrophages (foam cells) within the arterial intima is a hallmark of plaque initiation and progression. Our data show an apparent dose-dependent increase in cholesterol efflux to Apo-A1 following treatment with OOPEs, with EVOOPE+ exhibiting superior efficacy compared to the standard EVOOPE (Figure 9A,B). These results highlight the importance of phenolic concentration in regulating lipid homeostasis within the macrophage compartment.

Among the individual compounds tested, HT showed a greater capacity to stimulate cholesterol efflux than Tyr (Figure 9C,D), consistent with its higher antioxidant potential and its known regulatory influence on lipid transporters such as ATP-binding cassette transporter A1 (ABCA1) and G1 (ABCG1). These transporters are essential for cholesterol export to Apo-A1 and HDL, respectively, and are under the transcriptional control of liver X receptors (LXRs). Phenolic compounds like HT have been reported to activate LXR-dependent transcription, thereby enhancing the expression of genes involved in RCT and mitigating lipid accumulation in foam cells [25]. These observations align with previous studies by Berrougui et al., who demonstrated increased ABCA1 protein levels and cholesterol efflux in macrophages exposed to phenolic-rich olive oil extracts, suggesting a conserved mechanism across models [26].

The enhanced efficacy of EVOOPE+ relative to the standard extract can likely be attributed to its higher content and diversity of phenolic compounds. While HT and Tyr are key contributors, EVOOPE+ also contains other structurally diverse phenolics, which may act synergistically to modulate lipid metabolism. Several of these compounds have demonstrated the ability to influence transcription factors involved in lipid and inflammatory signaling (e.g., PPARγ, LXRα) [25], further supporting their integrated role in cholesterol homeostasis. Moreover, by reducing oxidative stress and inflammation, both of which impair cholesterol efflux capacity, olive oil phenolics indirectly preserve the functional integrity of RCT-related pathways.

Nevertheless, despite the promising in vitro evidence and mechanistic plausibility, preclinical data on the in vivo efficacy of individual phenolics remain mixed. For example, López de las Hazas et al. reported that HT supplementation in mice was associated with features of systemic dyslipidemia, raising questions about potential off-target effects or dose thresholds [27]. Similarly, Acín et al. observed no improvements in HDL-related markers and even reported an increase in lesion size and monocyte activation in apoE-deficient mice treated with olive phenolics [28]. In contrast, studies in hyperlipidemic rabbits have shown that HT supplementation improves lipid profiles and significantly reduces aortic lesions [29]. These discrepancies may reflect interspecies differences in metabolism, differences in experimental diets, the gut microbiota’s role in phenolic biotransformation, or variations in bioavailability and tissue distribution. It is also worth noting that the phenolic-mediated enhancement in cholesterol efflux is not purely a lipid-modulatory action but may also contribute to the resolution of inflammation within atherosclerotic lesions. Efflux of cholesterol from macrophages is closely tied to their phenotypic polarization; cholesterol-depleted macrophages are more likely to adopt an anti-inflammatory, M2-like profile, which supports plaque stabilization and tissue repair [3]. In this regard, the dual anti-inflammatory and pro-efflux activities of OOPCs may act in concert to reduce atherogenic burden. Taken together, the results presented here reinforce the atheroprotective role of OOPCs by highlighting their capacity to enhance cholesterol efflux from macrophages, likely via upregulation of ABCA1/G1 and modulation of LXR-related pathways. While HT emerges as a potent effector, the superior performance of EVOOPE+ suggests that a full spectrum of olive-derived phenolics may be necessary to achieve optimal cardioprotective effects. These findings contribute to the growing body of evidence supporting the integration of phenolic-rich EVOO in cardiovascular health strategies, warranting further investigation to define optimal dosages, combinations, and clinical applications [21].

## 4. Materials and Methods

### 4.1. Chemicals and Reagents

Tyrosol, Tert butyl hydroperoxide (TBHP), 2-thiobarbituric acid (TBA), hydrochloric acid (HCL), butylated hydroxytoluene (BHT), radioimmunoprecipitation assay buffer (RIPA), dimethyl sulfoxide (DMSO), and 3-(4,5-dimethylthiazolyl-2)-2,5-diphenyltetrazolium bromide (MTT) were purchased from Sigma-Aldrich (St. Louis, MO, USA), except for lipopolysaccharide (LPS), which were from Sigma-Aldrich Ltd. (Oakville, ON, Canada). Trichloroacetic acid (TCA) and butanol were from Fisher Scientific (Loughbor ough, UK). Dichlorodihydrofluorescein diacetate (H2-DCFH-DA) was purchased from Invitrogen (Waltham, MA, USA). Pierce BCA protein assay kit was from Thermo Fisher Scientific (Waltham, MA, USA). Ethanol, phorbol myristate acetate (PMA), and bovine serum albumin (BSA) were from Wisent Inc. (Saint-Jean-Baptiste, QC, Canada). Hydroxytyrosol was from ApexBio (Houston, TX, USA).

### 4.2. Plant Material

Two samples of olive oil were used in this study: a high-phenolic content extra-virgin olive oil (EVOO+) and an extra-virgin olive oil (EVOO), both obtained from the Atlas Olive Oil company (Casablanca, Morocco). 

### 4.3. Extraction of the Phenolic Fraction

The phenolic fraction was extracted using the procedure described by Pirisi et al. Briefly, olive oil was initially mixed with n-hexane, and then an 80% (methanol:water) solution was added. The resulting mixture was separated via centrifugation, and the phenolic-rich layer was collected, dried under vacuum, then lyophilised and stored at −20 °C until use [30].

### 4.4. Total Phenolic Content

The total phenolic content was measured using the Folin–Ciocalteu method [31]. Phenolic extracts were mixed with Folin–Ciocalteu reagent and Na_2_CO_3_, incubated in the dark for 2 h, and the absorbance was recorded at 760 nm. Quantification was based on a gallic acid calibration curve, and the results were expressed as mg GAE/kg oil (Appendix A).

### 4.5. Cell Culture

The J774 (murine macrophages) and human THP-1 monocytes were purchased from the American Type Culture Collection (Manassas, VA, USA) via Cederlane^®^ company (Burlington, ON, Canada). J774 cells were cultured in Dulbecco’s Modified Eagle Medium (DMEM). THP-1 cells were maintained in Roswell Park Memorial Institute (RPMI) medium. For differentiation into macrophages, THP-1 cells were treated with PMA. All mediums were supplemented with 10% fetal bovine serum, 10 U/mL penicillin, and 10 µg/mL of streptomycin and maintained in a 5% CO_2_ humidified incubator at 37 °C.

### 4.6. Cell Viability Measurement

The MTT assay assesses cell viability by measuring the reduction in a soluble yellow tetrazolium salt (MTT) in metabolically active cells via a mitochondria-dependent reaction, which leads to the formation of insoluble purple formazan crystals. The MTT viability test was performed using the method described by Denizo et al. [32]. Briefly, THP-1 macrophages were seeded in 96-well plates (10^4^ cells/well) and incubated for 24 h with olive oil phenolic extracts (OOPEs), Tyr, or HT. The medium containing the treatments was removed, and then 200 µL of 0.5 mg/mL MTT, prepared in fresh medium, was added to each well. The plate was then incubated for 4 h at 37 °C. After incubation, the MTT-containing medium was removed, and the formazan formed was solubilised in DMSO. Afterwards, absorbance was measured at 570 nm using a microplate reader (PerkinElmer, Woodbridge, ON, Canada). The results were expressed as percentage of cell viability and calculated using the following equation:$$\mathrm{Cell}\mathrm{Viability}\left(\%\right)=\left[\frac{A_{sample}}{A_{control}}\right]\times 100$$
A_sample_ is the absorbance of treated cells, and A_control_ is the absorbance of untreated cells.

### 4.7. Measurement of Intracellular ROS

Intracellular ROS levels were assessed using the fluorescent probe H2-DCFH-DA [33]. Briefly, J774 cells were seeded in a 12-well plate with complete DMEM medium and pretreated with OOPEs (25–150 µg/mL), Tyr, or HT (25–150 µM). Following pretreatment, cells were stimulated with LPS (1 µg/mL) for 24 h, then washed and exposed to DCFH-DA (10 µM) for 40 min. After incubation, the probe was then removed, and cells were washed before fluorescence intensity was measured using a VICTOR Multilabel Plate Reader (PerkinElmer, Canada) at excitation and emission wavelengths of 485 nm and 530 nm, respectively. Protein content was determined using the BCA assay, and ROS levels were normalized accordingly. Data were expressed as fluorescence intensity per milligram of protein for each experimental condition.

### 4.8. Lipid Peroxidation Assay

MDA levels in the supernatants from treated cells were measured with the TBARS test as previously described [34]. J774 cells were pretreated with OOPEs (25–150 µg/mL), Tyr or HT (25–150 µM) for 24 h, followed by exposure to 200 µM tert-butyl hydroperoxide (TBHP) for 2 h to induce oxidative stress. After treatment, the supernatant was collected by centrifuging the cell suspension. For the determination of malondialdehyde (MDA) levels, 300 µL of cell-free supernatant was combined with a previously prepared mixture containing 0.5% TBA, 30% TCA, 0.33M HCl, and 0.005% BHT. The reaction mixture was incubated in a boiling water bath for 45 min, then cooled for 10 min. Subsequently, 300 µL of butanol was added to each tube to extract the MDA-TBA adduct, followed by vortexing and centrifugation to separate the phases. The upper butanol phase was transferred to a 96-clear-bottom black plate well, and fluorescence was measured using a VICTOR Multilabel Plate Reader (PerkinElmer, Canada) at an excitation wavelength of 530 nm and an emission wavelength of 590 nm. MDA concentrations were normalized to protein content, determined using the (BCA) assay.

### 4.9. M1/M2 Polarization of THP1 Macrophages

In this assay, we evaluated the expression of the cell surface receptors CD86 and CD163, which are associated with M1 and M2 phenotypes, respectively. Cells were stimulated with LPS (1 µg/mL) and simultaneously treated with increasing concentrations of olive oil phenolic extracts (OOPEs) (25–150 µg/mL), Tyr or HT (25–150 µM) for 24 h. After treatment, cells were collected, washed twice with cold PBS, and centrifuged at 350× g for 6 min at 4 °C, then labeled with FITC-conjugated anti-CD86 and BV711-conjugated anti-CD163 monoclonal antibodies (BD Biosciences, Franklin Lakes, NJ, USA), then incubated at 4 °C for 50 min in the dark. After labelling, cells were washed, centrifuged again and resuspended in cold PBS before analysis by flow cytometry. Data were acquired using a CytoFLEX cytometer (Beckman Coulter, Brea, CA, USA) and processed using FlowJo 10.2 software (Tree Star Inc., Ashland, OR, USA).

### 4.10. Intracellular Inflammatory Markers Measurement

THP-1 macrophages were stimulated with LPS (1 µg/mL) and simultaneously treated with increasing concentrations of OOPEs (25–150 µg/mL), Tyr or HT (25–150 µM). Brefeldin A (1:1000; Abcam, Toronto, ON, Canada) was added during the last 4 h of incubation. Cells were then fixed, permeabilized with Fix/Perm buffer (BD Biosciences) for 50 min at 4 °C in the dark and stained with monoclonal antibodies against IL-10 (APC-rat anti-human), IL-1β (PE-mouse anti-human), IL-6 (BV421-mouse anti-human), IFN-α (PE-mouse anti-human) or NLRP3 (Alexa 647-mouse anti-human) (BD Biosciences) for 50 min at 4 °C in the dark. Data were acquired on a CytoFLEX cytometer (Beckman Coulter, Brea, CA, USA) and analyzed using FlowJo 10.2 (Tree Star Inc., Ashland, OR, USA).

### 4.11. Cholesterol Efflux Measurement in J774 Macrophages

Cholesterol Efflux in J774 Macrophages was measured as previously described with slight modifications [26]. Briefly, J774 macrophages were seeded at 1 × 10^5^ cells/well in 12-well plates and incubated overnight at 37 °C, 5% CO_2_. Cholesterol loading was performed by replacing the medium with serum-free DMEM containing 0.5% BSA and [^3^H]-cholesterol (1 μCi/mL) for 24 h, followed by three washes with PBS. Cells were then treated overnight with OOPEs (25–150 µg/mL), Tyr or HT (25–150 µM) in serum-free DMEM, with a positive control of cAMP (0.3 mM). After washing, cells were incubated with ApoA-I (25 μg/mL) for 5 h to induce ABCA1-mediated efflux. Supernatants were collected, and cells were lysed with 0.1 M NaOH. Radioactivity in the supernatant (^3^H-cholesterol efflux) and lysates (retained ^3^H-cholesterol) was measured using a scintillation counter.

### 4.12. Statistical Analysis

Statistical analysis was performed using GraphPad Prism version 9.0.0 (GraphPad Software, Inc., La Jolla, CA, USA), and the results are presented as mean ± SEM. Group differences were assessed using a *t*-test for two-group comparisons or one-way ANOVA followed by a multiple comparison test. Statistical significance was set at * *p* < 0.05, ** *p* < 0.01, *** *p* < 0.001.

## 5. Conclusions

This study provides compelling evidence for the multifaceted atheroprotective properties of olive oil polyphenols, demonstrating significant antioxidant, anti-inflammatory, and anti-atherogenic activities. Specifically, these compounds effectively mitigated intracellular oxidative stress, modulated key inflammatory mediators, and induced a macrophage phenotype associated with inflammation resolution and vascular protection. Furthermore, they promoted cholesterol efflux, a key process in reverse cholesterol transport, thereby directly supporting their role in mitigating atherogenesis. A critical observation was the biphasic, dose-dependent nature of these responses; at higher concentrations, the beneficial effects were attenuated or reversed, suggesting a potential shift towards pro-oxidant behavior or the activation of compensatory mechanisms. This emphasizes the need for careful dose optimization when considering their therapeutic or nutraceutical applications. Overall, our study reinforces the health-promoting potential of polyphenol-rich EVOO and highlights the critical importance of dose in maximizing its cardiometabolic benefits.

## Figures and Tables

**Figure 1 ijms-26-11165-f001:**
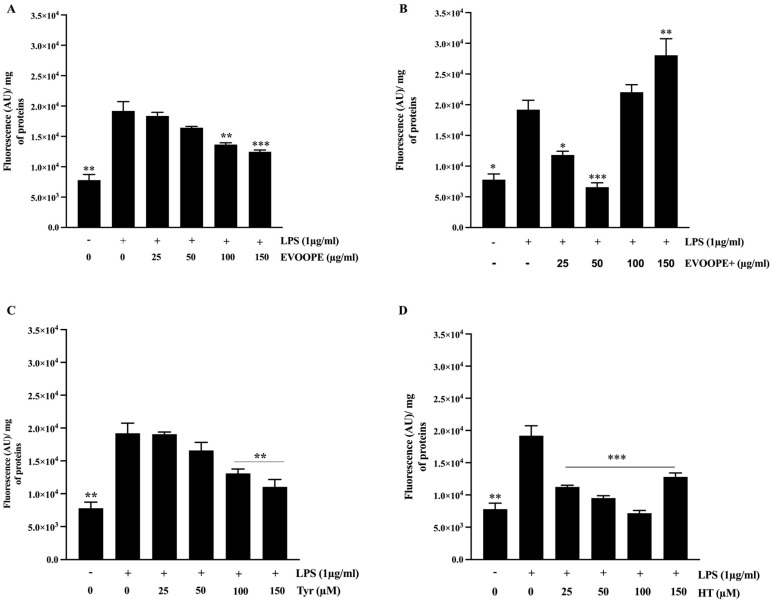
EVOO polyphenols reduce ROS generation in J774 macrophages. Cells were stimulated with LPS (1 µg/mL) for 24 h. Macrophages were pretreated with EVOOPE (**A**) and EVOOPE+ (**B**) extracts, Tyr (**C**), and HT (**D**) before their stimulation with LPS (1 µg/mL). The data are presented as mean ± SEM of at least three independent experiments. (* *p* < 0.05; ** *p* < 0.01; *** *p* < 0.001. vs. LPS).

**Figure 2 ijms-26-11165-f002:**
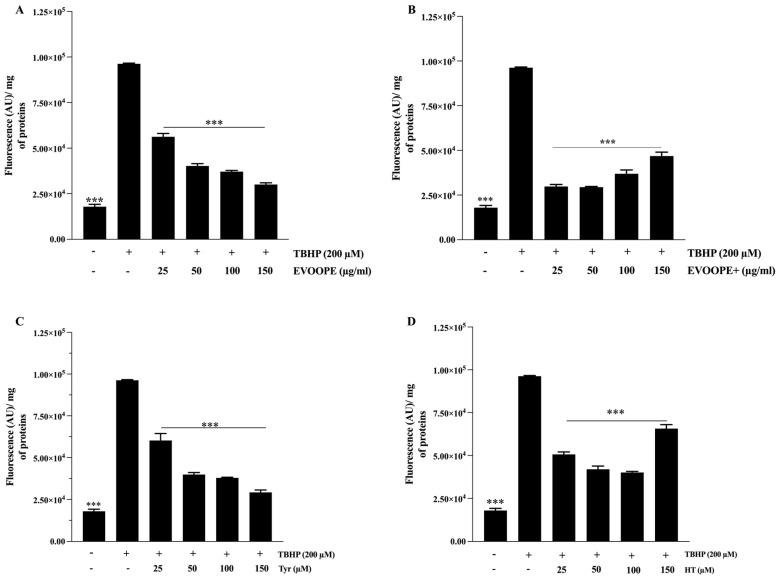
The protective effect of polyphenol extracts on lipid peroxidation. J774 macrophages were pretreated (24 h) with EVOOPE (**A**) and EVOOPE+ (**B**) extracts, Tyr (**C**), and HT (**D**) before their stimulation with TBHP (200 µM). The data are presented as mean ± SEM of at least three independent experiments. (*** *p* < 0.001. vs. TBHP).

**Figure 3 ijms-26-11165-f003:**
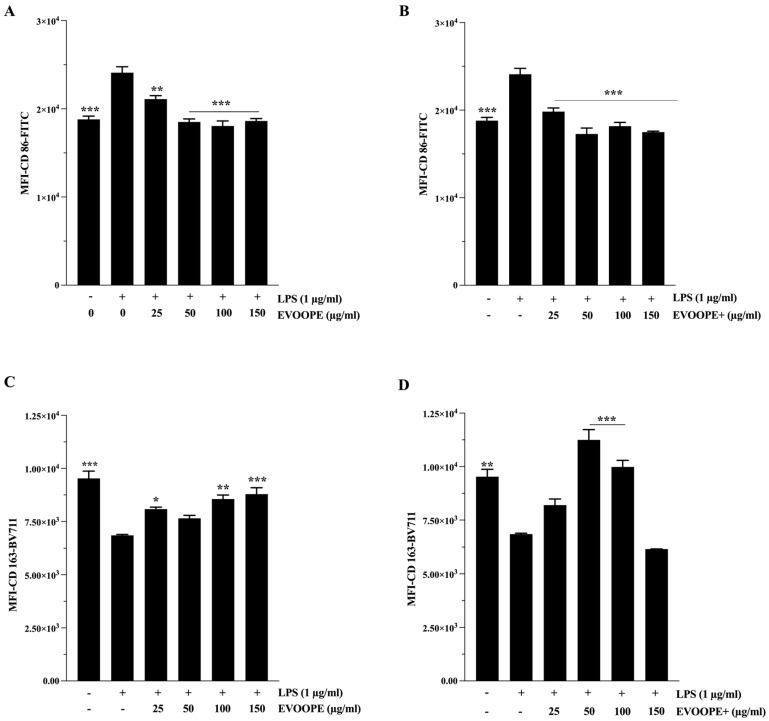
The modulatory effects of olive oil polyphenols on M1/M2 polarization THP-1 derived macrophages. THP-1 cells were stimulated by (1 µg/mL) of LPS and cotreated simultaneously with polyphenols for 24 h. (**A**,**B**) represents the effect of EVOOPE and EVOOPE+ on CD86 expression. (**C**,**D**) illustrates the effect of EVOOPE and EVOOPE+ on CD163 expression. The data are presented as mean ± SEM of at least three independent experiments. (* *p* < 0.05; ** *p* < 0.01; *** *p* < 0.001. * vs. LPS).

**Figure 4 ijms-26-11165-f004:**
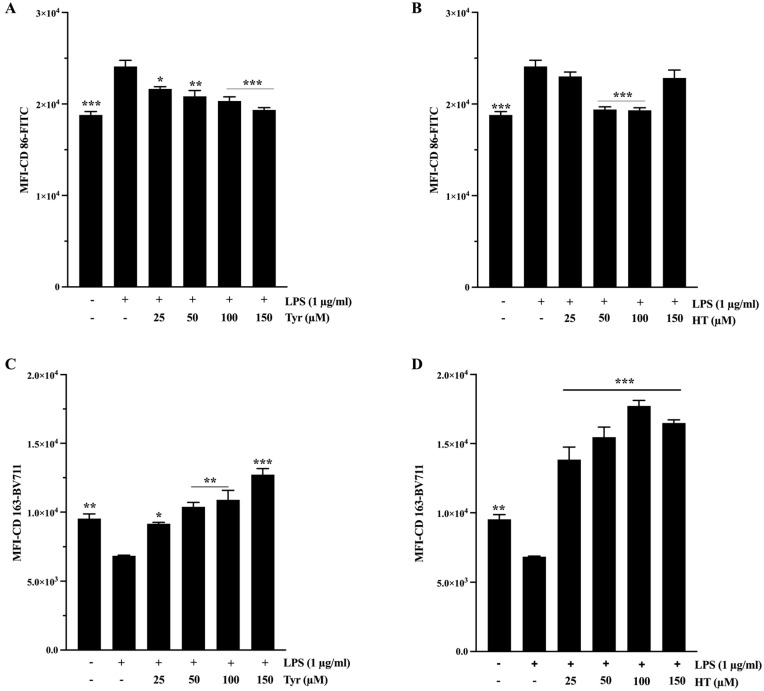
The modulatory effects of Tyr and HT on M1/M2 polarisation of THP-1-derived macrophages. Cells were stimulated by (1 µg/mL) of LPS and cotreated simultaneously with polyphenols for 24 h. (**A**,**B**) represents the effect of Tyr and HT on CD86 expression. (**C**,**D**) illustrates the effect of Tyr and HT on CD163 expression. The data are presented as mean ± SEM of at least three independent experiments. (* *p* < 0.05; ** *p* < 0.01; *** *p* < 0.001. vs. LPS).

**Figure 5 ijms-26-11165-f005:**
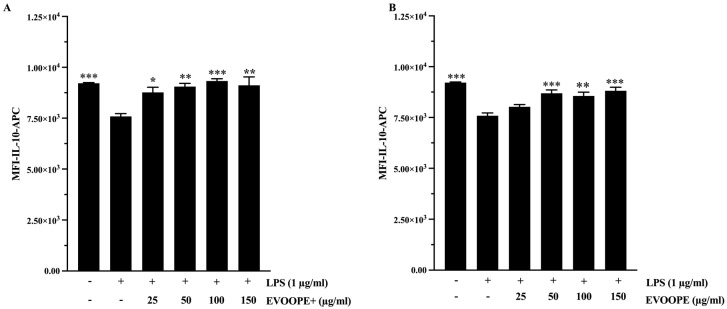
The bioeffects of EVOOPE and EVOOPE+ on gene expression of IL-10. The THP-1 human-derived macrophages were stimulated with 1 (µg/mL) of LPS and cotreated simultaneously overnight with EVOOPE+ (**A**), EVOOPE (**B**) at different concentrations. The data are presented as mean ± SEM of at least three independent experiments. (* *p* < 0.05; ** *p* < 0.01; *** *p* < 0.001. vs. LPS).

**Figure 6 ijms-26-11165-f006:**
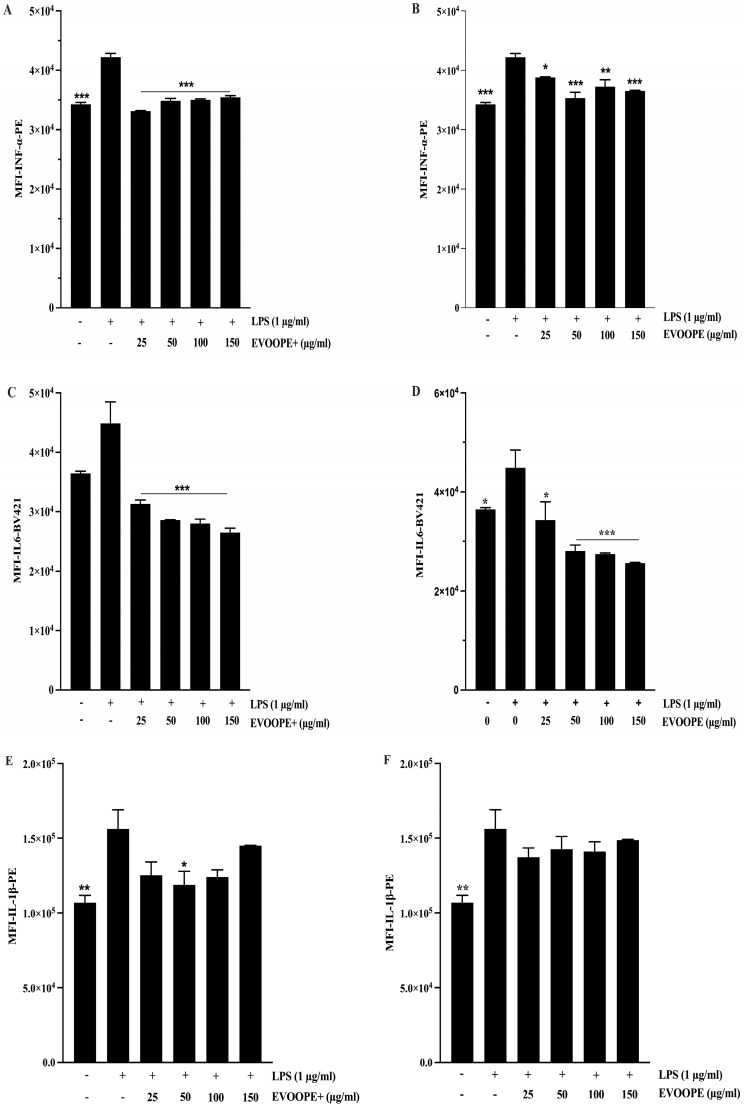
Effect of EVOOPE+ and EVOOPE on the expression of proinflammatory markers. The THP-1 human-derived macrophages were stimulated with 1 (µg/mL) of LPS and cotreated simultaneously for 24 h with EVOOPE+ ((**A**,**C**,**E**); respectively for INF-α, IL-6 and IL-1β) and EVOOPE ((**B**,**D**,**F**); respectively for INF-α, IL-6 and IL-1β) at increased concentrations. The data are presented as mean ± SEM of at least three independent experiments. (* *p* < 0.05; ** *p* < 0.01; *** *p* < 0.001. vs. LPS).

**Figure 7 ijms-26-11165-f007:**
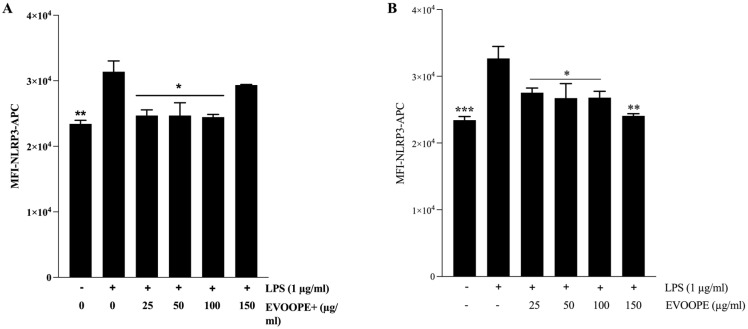
Effect of EVOOPE and EVOOPE+ on NLRP3 inflammasome expression. The THP-1 human-derived macrophages were stimulated with 1 µg/mL of LPS and cotreated simultaneously for 24 h with EVOOPE+ (**A**), EVOOPE (**B**) at different concentrations. The data are presented as mean ± SEM of at least three independent experiments. (* *p* < 0.05; ** *p* < 0.01; *** *p* < 0.001. vs. LPS).

**Figure 8 ijms-26-11165-f008:**
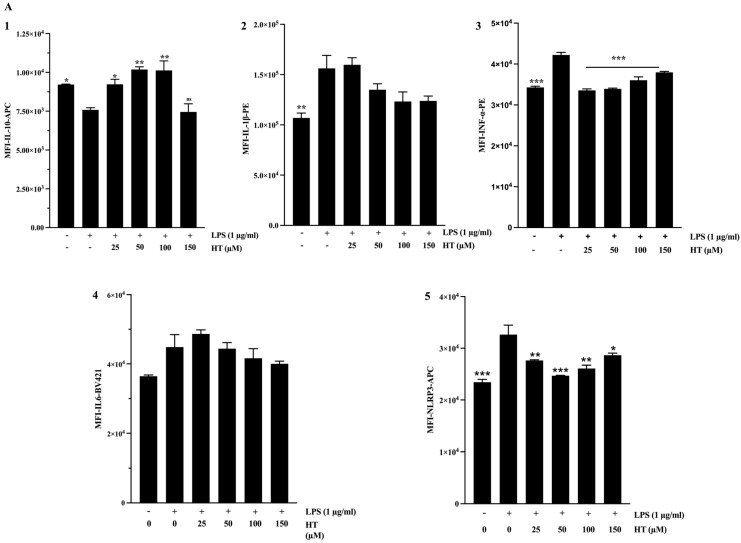
Effect of Tyr and HT on the expression of proinflammatory markers. The THP-1 human-derived macrophages were stimulated with (1 µg/mL) of LPS and cotreated simultaneously for 24h with HT ((**A1**–**A5**); respectively for IL-10, IL-1β, INF-α IL-6 and NLRP3) and Tyr ((**B1**–**B5**); respectively for IL-10, IL-1β, INF-α IL-6 and NLRP3) at increased concentrations. The data are presented as mean ± SEM of at least three independent experiments. (* *p* < 0.05; ** *p* < 0.01; *** *p* < 0.001. vs. LPS).

**Figure 9 ijms-26-11165-f009:**
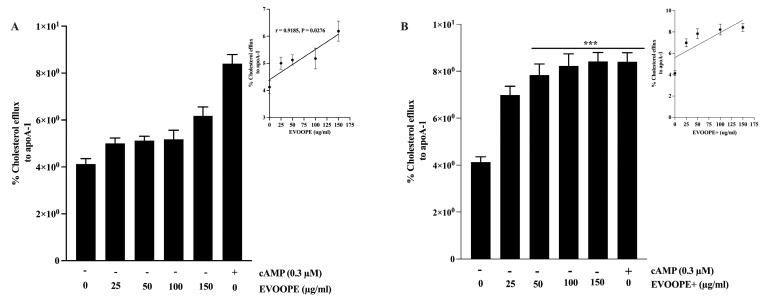
[^3^H]-cholesterol–loaded J774 macrophages were treated overnight with (**A**) EVOOPE+, (**B**) EVOOPE, (**C**) Tyr, (**D**) HT, or cAMP (0.3 mM), followed by incubation with ApoA-I (25 μg/mL) for 5 h to assess ABCA1-mediated cholesterol efflux. Data represent mean ± SEM of ≥3 independent experiments. (* *p* < 0.05, *** *p* < 0.001. vs. Control).

**Table 1 ijms-26-11165-t001:** Quantification of total phenolic compounds.

Extract	TPC (mg GAE/Kg of Oil)
EVOOPE+	1525.13 ± 21.21 ***
EVOOPE	715.6 ± 8.61

The results are expressed as mean ± SEM of at least three independent experiments. A *t*-test was used to compare the total phenolic content of the olive oil extracts. (*** *p* < 0.001).

**Table 2 ijms-26-11165-t002:** Effect of Treatments on Cell Viability. THP-1 macrophages were exposed to increasing concentrations of EVOOPE+, EVOOPE, Tyr (tyrosol), or HT (hydroxytyrosol) for 24 h to assess their impact on cell viability using the MTT reduction assay.

	0 (µg/mL)	25 (µg/mL)	50 (µg/mL)	100 (µg/mL)	150 (µg/mL)	25 (µM)	50 (µM)	100 (µM)	150 (µM)
Control	100 ± 2.70%	-	-	-	-	-	-	-	-
EVOOPE+	-	129.68 ± 4.25% ***	122.36 ± 0.17% **	107.85 ± 1.93%	78.77 ± 2.14% **	-	-	-	-
EVOOPE	-	117.64 ± 2.37%	107.69 ± 1.68%	112.23 ± 2.92%	121.28 ± 3.37% **	-	-	-	-
HT	-	-	-	-	-	120.07 ± 1.76% ***	125.72 ± 0.37% ***	115.18 ± 1.89% **	99.13 ± 1.47%
Tyr	-	-	-	-	-	121.28 ± 3.37% **	117.47 ± 2.06% *	109.29 ± 1.62%	106.83 ± 3%

The data are expressed as mean ± SEM of at least three independent experiments. Statistical analysis was carried out with one-way ANOVA followed by a Tukey post hoc test for multiple comparisons (* *p* < 0.05; ** *p* < 0.01; *** *p* < 0.001. vs. control).

## Data Availability

The original contributions presented in this study are included in the article/Appendix A. Further inquiries can be directed to the corresponding author.

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
