# Peer review of "Biological Activities Underlying the Cardiovascular Benefits of Olive Oil Polyphenols: Focus on Antioxidant, Anti-Inflammatory, and Anti-Atherogenic Effects"

_ijms, 2025, doi:10.3390/ijms262211165_

Round 1
Reviewer 1 Report
Comments and Suggestions for Authors# The manuscript which title '' Molecular Mechanisms Underlying the Cardiovascular Benefits of Olive Oil Polyphenols: Focus on Antioxidant, Anti-Inflammatory, and Anti-Atherogenic Effects” is good but need more improvements as follow:
#The introduction needs more improvements it should have more details about the chemical composition of EVOO and clear the difference between.
# The material and methods section need mention the method of GC/MS or LC- Mass to explain the chemical composition of two olive oils under the study.
#The results section should improve:
Table 2. Effect of Treatments on Cell Viability needs more explanation in the results and mention the cell viability equation and check the data in the table as well as why the concentrations are different in units??
# All figures need to put in high resolution not clear and not valid to publish with its state now.
# The paper without the LC /MS or GC/MS to fatty acids and compare the difference between is not valid to publish where is the novelty???
Comments on the Quality of English Language# The quality of the language and gamer need more improvements.
Author Response
Comment 1: (The introduction needs more improvements; it should have more details about the chemical composition of EVOO and clarify the differences between classes of olive oils”)
Response 1: The introduction has been revised to include a detailed overview of the chemical composition of EVOO, Additionally, we clarified the differences between extra virgin olive oil, virgin olive oil, and refined olive oil, highlighting distinctions in extraction method, acidity, phenolic content, and bioactive properties.
Comment 2: (The materials and methods section needs to mention the method of GC/MS or LC-MS to explain the chemical composition of the two olive oils under study)
Response 2: We thank the reviewer for this suggestion. We understand that the GC/MS or LC-MS techniques can be a powerful tool for chemical characterisation of the used samples. Effectively, we already did this characterisation. For LC-MS analysis, we refer the reviewer to our previously published article (DOI: 10.3390/antiox13010130).
Fatty acids analysis was performed using GC-FID (Agilent Technologies) by the biotechnology analytical platform at the Research Centre on Ageing (University of Sherbrooke, QC, Canada). Results are attached to the current revision (Supplementary material Table S1).
Comment 3: (Table 2. Effect of Treatments on Cell Viability needs more explanation in the results and mention the cell viability equation and check the data in the table as well as why the concentrations are different in units) ??) Response 3: We thank the reviewer for this observation. We have revised the Results section to provide a clear description of Table 2. Cell viability was calculated using the standard MTT assay equation:
Cell viability (%) = (Asample/AControl)*100
This equation is included in the Methods section of the current version. Concentrations are expressed in both µg/mL (for polyphenol extracts) and µM (for pure compounds, HT and Tyr). The chosen units and concentration ranges were based on previously published studies to ensure consistency and comparability with the literature.
Comment 4: (All figures need to be in high resolution; current figures are not clear and not valid for publication) Response 4: We have replaced all figures with high-resolution versions suitable for publication, ensuring clear visibility of all labels, legends, and data points. Comment 5: The paper without LC-MS or GC-MS analysis of fatty acids and comparison of the differences between oils is not valid to publish; where is the novelty? Response 5: We agree with the reviewer that these techniques could bring an important contribution to our paper. Indeed, as we respond to comment 2, we have already conducted this analysis. For GC, please see the attached file in the supplementary materials. For LC-MS, we refer the reviewer to our published research (DOI: 10.3390/antiox13010130). Comment 6: (The quality of the language needs improvement) Response 6: We thank the reviewer for the comment. Authors have thoroughly revised the manuscript for clarity, grammar, and scientific style. All sections have been carefully proofread, and any ambiguous expressions have been clarified.
Reviewer 2 Report
Comments and Suggestions for Authors
The manuscript titled “Molecular Mechanisms Underlying the Cardiovascular Benefits of Olive Oil Polyphenols: Focus on Antioxidant, Anti-Inflammatory, and Anti-Atherogenic Effects” is devoted to a detailed study of various types of biological activity of olive oil and phenolic compounds extracted from it. MTT, intracellular ROS level, lipid peroxidation assay, impact on macrophages methods, anti-inflammatory activity measurements are used. Of course, it is important to understand the potential benefits of plant-based products, so the relevance of the study is quite high. At the same time, the work is a somewhat superficial study, being limited for the most part to some description of experimental data. One supplier of olive oil samples is used in the work. The issue of the chemical composition of the oil used remains unclear, which means that it is difficult to assess the possibility of generalizing the conclusions reached. The “Molecular Mechanisms” stated in the title of the article suggest a more detailed description of the action of polyphenols than a demonstration of the well-known fact of their antioxidant effect.
The work may be of interest to experts in the field of food technology and phytochemistry. However, it requires a more thorough study.
I believe that the manuscript can be published in the International Journal of Molecular Sciences only after major revision after taking into account general comment and a few specific comments:
- The introduction section is quite short, and it does not present some valuable information. The term "polyphenols" should be clearly defined, and the components identified in olive oil need to be mentioned. Additionally, in this section you need to clearly state the aim of your work.
- In the work, the authors claim that the main polyphenols contained in olive oil are tyrosol and hydroxytyrosol. Data on the bioactivity of olive oil were compared with those for tyrosol and hydroxytyrosol. However, it is not clear what the content of these compounds is in the olive oil samples used in this work.
- Section 4.6. name: an extra period has been added.
- Calibration curve parameters should be presented in the Section 4.4.
Author Response
Comment 1: “The introduction section is quite short, and it does not present some valuable information. The term "polyphenols" should be clearly defined, and the components identified in olive oil need to be mentioned. Additionally, in this section you need to clearly state the aim of your work.”
Response 1: We appreciate the reviewer’s feedback regarding the introduction section. To better introduce the content of our paper, we have expanded the introduction for a clear definition of polyphenols. The major phenolic compounds in olive oil are now described. Additionally, we have clearly stated the aim of the study.
Comment 2: “In the work, the authors claim that the main polyphenols contained in olive oil are tyrosol and hydroxytyrosol. Data on the bioactivity of olive oil were compared with those for tyrosol and hydroxytyrosol. However, it is not clear what the content of these compounds is in the olive oil samples used in this work.”
Response 2: We appreciate the reviewer’s observation and agree that presenting the content of these compounds is essential. In this revised version, we have added quantitative data on the content of HT and Tyr in both EVOO+ and EVOO in the Results section.
Comment 3: “Section 4.6. name: an extra period has been added.”
Response 3: We thank the reviewer, and we apologize for the typographical error. We corrected the typo as suggested.
Round 2
Reviewer 1 Report
Comments and Suggestions for Authors
Thanks for your response but the manuscript needs some improvements in some sections like results please, follow my comments in the attached manuscript PDF.

Author Response
Authors’ response to reviewers-Second round
Manuscript Number: ijms-3918226
Manuscript Title: Molecular Mechanisms Underlying the Cardiovascular Benefits of Olive Oil Polyphenols: Focus on Antioxidant, Anti-Inflammatory, and Anti-Atherogenic Effects
Authors: Boumezough et al., 2025
Response to reviewer 1:
Comment 1: “add abbreviation section and add all abbreviation of expressions before introduction or before references according to the instructions of the journal”
Author’s response: We think the reviewer for this suggestion. All the used abbreviations were added to the revised version.
Comment 2: “with concentrations of 233.6 mg/kg …. do you mean mg GAE/Kg oil ?????”
Author’s response: We thank the reviewer for this observation. The mentioned values referred to the concentration of Tyrosol and Hydroxytyrosol in each olive oil (mg/kg of oil). Only the total phenolic compounds are expressed in (mg GAE/Kg of oil), as mentioned in the Table 2.
Comment 3: “write statistical analysis under the table 1??”
Author’s response: We appreciate the reviewer's observation on this detail, and it has been addressed in the revised version.
Comment 4: “Table 2. Where is the statistical analysis?? Write it in the table and under the table”
Author’s response: The statistical analyses have been added as requested under Table 2.
Comment 5: “Determine the IC50 of Olive oils concerning HT and Tyr”
Author’s response: We appreciate the reviewer’s comment regarding the calculation of ECâ‚…â‚€ values. However, in our case, the response to the treatments did not follow a monotonic dose–response pattern. In the absence of such a relationship, the determination of an ECâ‚…â‚€ would not provide a reliable or biologically meaningful estimate, as this parameter assumes a sigmoidal, continuous progression of the effect with increasing concentrations. For these reasons, and in accordance with previous publications using the TBARS assay and similar oxidative stress assays, we chose to present our results as fluorescence intensity normalized to protein content (AU/mg protein), which provides a more accurate and interpretable reflection of the experimental data.
Comment 6: “Figure 6 : Replace with high resolution”
Author’s: The resolution of the figure has been improved as requested
Comment 6: “Figure 9 : Replace with high resolution”
Author’s: The resolution of the figure has been improved as requested
Comment 7: “Table S1: Values are expressed as mean ± esm. do you mean SEM please, correct”
Author’s response: Yes, we mean SEM. We apologize for this error, and it has been corrected accordingly.
Reviewer 2 Report
Comments and Suggestions for Authors
I see that the authors have done a great job. The information in the presented version of the manuscript gives a more complete picture of the research.
However, I would like to draw your attention to the fact that the article is devoted more to the study of biological activity, rather than molecular mechanisms. I would recommend reformulating the title of the work and the goals, taking into account that it is the biological activity that is being studied, and not the molecular mechanisms.
Thus, I believe that the manuscript can be accepted for publication after the minor revision.
Author Response
Authors’ response to reviewers-Second round
Manuscript Number: ijms-3918226
Manuscript Title: Molecular Mechanisms Underlying the Cardiovascular Benefits of Olive Oil Polyphenols: Focus on Antioxidant, Anti-Inflammatory, and Anti-Atherogenic Effects
Authors: Boumezough et al., 2025
Response to reviewer 2:
“Comments and Suggestions for Authors I see that the authors have done a great job. The information in the presented version of the manuscript gives a more complete picture of the research. However, I would like to draw your attention to the fact that the article is devoted more to the study of biological activity, rather than molecular mechanisms. I would recommend reformulating the title of the work and the goals, taking into account that it is the biological activity that is being studied, and not the molecular mechanisms. Thus, I believe that the manuscript can be accepted for publication after the minor revision.”
Author’s response: We would like to thank the reviewer for the constructive and positive feedback. We appreciate the suggestion about the title. In response, we have completely revised the title and aim to reflect the bioactivities rather than the molecular mechanisms.
Here is what we propose as a title :
Biological Activities Underlying the Cardiovascular Benefits of Olive Oil Polyphenols: Focus on Antioxidant, Anti-Inflammatory, and Anti-Atherogenic Effects
Round 3
Reviewer 1 Report
Comments and Suggestions for Authors Thanks for the response of my comments.